# A review and upgrade of the Lithospheric dynamics in context of the Seismo-electromagnetic Theory

Patricio Venegas-Aravena (1, 2, 3), Enrique G. Cordaro (1, 4), David Laroze (5).

(1) Cosmic Radiation Observatories, University of Chile, Casilla 487-3, Santiago, Chile
(2) Departamento de Geofísica, Universidad de Chile, Blanco Encalada 2002, Santiago, Chile.
(3) Department of Structural and Geotechnical Engineering, School of Engineering, Pontificia Universidad Católica de Chile, Vicuña Mackenna 4860, Macul, Santiago, Chile.
(4) Facultad de Ingeniería, Universidad Autónoma de Chile, Pedro de Valdivia 425, Santiago, Chile.
(5) Instituto de Alta Investigación, CEDENNA, Universidad de Tarapacá, Casilla 7D, Arica, Chile.

Author: P. Venegas-Aravena: patricio.venegas@ing.uchile.cl
Co-author: E. G. Cordaro: ecordaro@dfi.uchile.cl
Co-author: D. Laroze: dlarozen@uta.cl

**Abstract**

This publication highlights theoretical work that could explain five different empirical observations indicating a direct relationship between magnetic fields and earthquakes, which would allow the description of a causal mechanism prior to and during the occurrence of earthquakes. These theoretical calculations seek to elucidate the role of the magnetic field in different aspects of solid earth dynamics, with an interest in the study and comprehension of the physics that could generate earthquakes accompanied by simultaneous magnetic signals within the lithosphere. The Motion of Charged Edge Dislocations (MCD) model and its correlation with the magnetic field have been used in order to include the generation of electric currents. The electric currents resulting from stress variation in the lithosphere helps us to analyze the lithosphere as a critical system, before and after the occurrence of earthquakes, by using the concept of earthquake entropy. Where it is found that the non-existence of seismic and magnetic precursors could be interpreted as a violation to the second law of thermodynamics. In addition, the Seismic Moment and the Moment Magnitude of some great earthquakes are quite accurately calculated using the co-seismic magnetic field. The distance-dependent co-seismic magnetic field has been theorized for some of the largest recorded earthquakes. The frequency of oscillation of the Earth's magnetic field that could be associated with earthquakes is calculated and being consistent to the ultra-low frequency (ULF) signals that some authors propose in the so-called "LAIC Effect" (lithosphere-atmosphere-ionosphere coupling). Finally, the location and dimensions of the micro cracks that explain some anomalous magnetic measurements are shown.

Keywords: Seismo-electromagnetic Theory, LAIC Effect, Magnetism, Earthquakes.

## 1 Introduction

A number of investigations attempting to relate the magnetic field to seismic events have emerged over the past few years, (e.g. Park, 1996; Surkov et al., 2003; Johnston et al., 2006; Balasis and Mandea 2007; Sgrigna et al., 2007; Saradjian and Akhoondzadeh, 2011; Varotsos et al., 2011; De Santis 2014; Donner et al., 2015; Schekotov and Hayakawa, 2015; Daneshvar and Freund, 2017; De Santis et al., 2017; Cordaro et al., 2018, 2019; Marchetti and Akhoondzadeh, 2018; Pulinets et al., 2018; among others). However,

there is still no unified causal mechanism that is widely accepted and that may account for the physics of all these observations prior to or during the occurrence of an earthquake (Hough, 2010), although the laboratory evidence shows the possibility of an increase in the conductivity of rocks when subjected to stress changes, either through microcracks or chemical imperfections (Freund, 2003; Anastasiadis et al., 2004; Cartwright-Taylor et al., 2014). Therefore, this paper will attempt to explain the physics of magnetic observations recorded by different researchers accurately, organizing them in five categories:

1.- Since the lithosphere can be considered a non-equilibrium system (De Santis et al., 2011), it is necessary to study any change in stress on rocks. The generation of current and magnetic field resulting from stress changes in rocks and their relationship with earthquakes has been shown empirically and theoretically by Vallianatos and Tzanis (2003), Anastasiadis et al. (2004), Scoville et al. (2015), among others. This information is relevant, as any mechanism to be related to earthquakes should provide some connection with stress changes in the lithosphere. Many explanations have been offered about the generation of currents through stress changes in rocks, including the piezoelectric effect (Tuck et al., 1977), the presence of fluids in rocks through the so-called electrokinetic effect (Morgan et al., 1989) or chemical processes in rocks (Paudel et al., 2018). However, the generation of transient currents occurs in rocks either with or without the presence of water or liquids (Yoshida et al., 1998), in non-piezoelectric materials (Freund and Borucki, 1999), and in materials under non-elastic conditions (Triantis et al., 2012). Thus, a simple model for the study of current generation by stress changes is the so-called Motion of Charged Edge Dislocations (MCD), which consists of the movement of charges due to the generation of microcracks within a brittle and semi-brittle material similar to the crust that has undergone a stress change (Triantis et al., 2012). Once the physical mechanism that generates magnetism by stress changes has been found, it is essential to study the temporal evolution of the lithospheric system, which is referred to in group 2.

2.- According to De Santis et al. (2011) and De Santis et al. (2014), the measurement of the temporal evolution of stress is achieved by measuring the "Earthquake Entropy", since the occurrence of an earthquake is an irreversible process comparable to a "critical system", due to the irreversible change in the state of such system, i.e. from a high-stress to a lower-stress lithosphere during an earthquake (De Santis et al., 2017). However, in order to correctly apply the stress configuration in an area of the lithosphere, it is necessary to know the "b-value" of Gutenberg-Richter's empirical law, since according to Schorlemmer et al. (2005), this value can be interpreted as a type of inverse measure of stress and therefore the temporal evolution of "b-value" could be related to the temporal evolution of stress and magnetic field through group 1.

3.- Once the evolution of the stress has been determined according to the magnetic field, the calculation of the Seismic Moment and the Moment Magnitude of Earthquakes will be carried out by using the co-seismic magnetic field since, as stated by Utada et al. (2011), a possible co-seismic magnetic variation of 0.8 nT was recorded at about 100 km from the Tohoku 2011 Mw9.0 earthquake rupture area while Johnston et al. (2006) also reported changes in the magnetic field close to earthquake fault during the Parkfield 2004 M6.0 earthquake and during the Loma Prieta 1989 Mw7.1 earthquakes also were reported possible co-seimic changes in magnetic field (Karakeliana et al., 2002).

4.- One of the most important group of measurements corresponds to the recording of ultra-low frequency (ULF) magnetic signals, i.e. frequencies below 1 Hz, as many researchers have found such anomalous frequencies prior to or during earthquake, mainly close to mHz and μHz (Fenoglio et al., 1995; Sorokin and Pokhotelov, 2010; Schekotov and Hayakawa, 2015; De Santis et al., 2017; Cordaro et al., 2018, 2019; Marchetti and Akhoondzadeh, 2018; among others), although according to Vallianatos and Tzanis (2003)

the magnetic field oscillation frequencies that could be related to earthquakes have a range of at least three orders of magnitude, so that kHz variations measured by other groups could also be included (Rozhnoi et al., 2008; Büyüksaraç et al., 2015; Potirakis et al., 2018a; among others).

5.- A final aspect to consider is the origin of the possible magnetic variations studied. The great problem of the LAIC effect is the lack of certainty about the mechanism that generates currents towards the atmosphere and ionosphere. Some authors consider that the currents are of external origin to the lithosphere (e.g. Marchetti and Akhoondzadeh, 2018), while others suggest internal origin (e.g. Vallianatos and Tzanis, 2003). To avoid this lack of consensus, it is essential to be able to define the approximate place where the currents are created and to explain the measurements of all the research groups during non-co-seismic times.

After the general description of each of these five topics, each theoretical framework is developed in sections 2, 3, 4, 5 and 6 respectively, maintaining the same order set out in this introduction. Finally, Section 7 summarizes the calculations and results obtained, and where the conclusions reached are presented.

**2 Rock physics, stress change, current generation and magnetic field**

The Zener-Stroh mechanism explains the generation and propagation of microcracks within a solid as the pile-up of edges dislocations at a certain location due critical external mechanical stress or load (e.g. Stroh, 1955). The movement of an edge dislocation stops when they encounter an obstacle or barrier within the solid (a scheme is shown in Figure 1a). Other edges dislocations may also reach the obstacle and will begin to pile up if they cannot overcome that obstacle (Figure 1b). This stacking will create a shear stress $\tau$, which will create a microcracks (blue triangle in Figure 1b) (e.g. Fan, 1994 and references therein). The microcracks can continue the propagation through different paths within the material (e.g. Xie and Sanderson, 1995) (blue lines in Figure 1c). This will generate avalanches of cracks due to the nucleation of neighboring cracks, which will allow large-scale cracks (blue lines in Figure 1d)(e.g. Main et al., 1993; Wang et al., 2015 and references therein).

For the other hands, the edges dislocations are electrically neutral in thermal equilibrium (Whitworth, 1975). However, the generation of microcracks is a dynamic process that breaks the ionic bonds that hold the solid together, so the microcracks will be accompanied by polarization and current density (e.g. Vallianatos and Tzanis, 1999). This phenomena is known as the Motion of Charged Edge Dislocations model (MCD model) (A scheme of polarization by MCD model is shown in Figure 1b, d). Several authors have shown that it is possible to detect electrification when a rock sample is compressed (Pressure Stimulating currents) uniaxially as shown in Figure 2a (e.g. Stavrakas et al., 2004 and references therein). It is thought that the electrification is due to the MCD model and it can scale with the rock fracture (Figure 1d) (e.g. Vallianatos and Triantis, 2008). According to Tzanis and Vallianatos (2002) the generation of a current density $J$ within rocks can be represented as the temporal change in plastic deformation that rocks undergo under compressional stress changes with time ($d\sigma/dt$) by:

$$J = \frac{\sqrt{2}}{\beta} \frac{q_l}{b} \left( \frac{1}{Y_{eff}} \frac{d\sigma}{dt} \right) \quad (1)$$

Where $q_l$ is the linear charge density of edge dislocation, $b$ is the Burgers vector module, $\beta$ varies between 1 and 1.5 and corresponds to the ratio $(\Lambda^+ + \Lambda^-)/(\Lambda^+ - \Lambda^-)$. $\Lambda^+$ and $\Lambda^-$ represent dislocations number created by compression and uniaxial tension within a rock (Whitworth, 1975; Vaillianatos and

Tzanis, 1998), and $Y_{eff}$ is the Young's effective module (Turcotte et al., 2003). Figure 2b is a schematic showing the direction of main currents $J$ when the stress $\sigma$ changes with time. The currents would tend to be parallel to the axes of fracture, however, the electrification of rocks can also propagate in other directions within the rock samples (Saltas et al., 2018) (Figure 1d).

On the other hand, Vallianatos and Tzanis (2003) model the magnetic field on the lithosphere surface as the magnetic field measured at the interface (with $r$ and $\theta$ coordinates) of a conductive half-space (since the rocks could become (semi)conductive when they undergo stress changes (Freund, 2003; Anastasiadis et al., 2004)). Then, the magnetic field could be created by a polarized sphere embedded in this conductive medium (Griffiths, 1996; Vallianatos and Tzanis, 2003). A scheme can be seen in Figure 3. According to Vallianatos and Tzanis, 2003, the magnetic field on the surface of the lithosphere is determined by:

$$\vec{B}(t) = \frac{3\mu_m V}{4\pi r^2} \sin\theta \frac{\partial P_2}{\partial t} \hat{z} \quad (2)$$

Where $\mu_m$ is the magnetic permeability of the medium (half-space), $J_2 = \frac{\partial P_2}{\partial t}$ is the horizontal current density, $r$ the distance to the sphere and the volume of the polarized sphere embedded in a medium. Equation 2 is valid for any source that generates polarization changes in the medium. According to Vallianatos and Tzanis (2003) if electric current is generated by microcracks then has a volume lower than $V$. This can be seen from the scheme of Figure 1d, where microcracks are represented by blue lines and do not cover the entire volume. The paths of these microcracks and their distribution are fractal in nature (e.g. Xie and Sanderson, 1995; Uritsky et al., 2004). According to Turcotte (1997), the fractal volume of all the microcracks within the medium can be represented by:

$$V \approx \frac{4\pi}{3} \frac{AD}{3-D} (l_{max})^{3-D} S_R \quad (3)$$

Where $l_{max}$ is the radius of the largest microcracks, $D$ is the rock fractal dimension, $S_R$ is a factor defined by $S_R = (1 - \left(\frac{l_{min}}{l_{max}}\right)^{3-D})$, where $l_{min}$ is the radius of the smallest microcrack. It is assumed that the ratio $\left(\frac{l_{min}}{l_{max}}\right)$ is small, so $S_R \approx 1$. The factor $A \approx (D-2)(l_{min})^{D-2} S$ appears from the fractal integration of the microcrack. Where $S$ is the largest fracture area. Therefore, the maximum magnetic field ($\sin\theta = \pi/2$) is reached by replacing Equation 3 in 2:

$$B \approx \frac{\mu_m AD}{(3-D)r^2} (l_{max})^{3-D} J_2 \quad (4)$$

If $J_2$ corresponds to the total current density $J$ present in the half-space, then Equation 1 may be replaced in 6:

$$B \approx \frac{\sqrt{2}q_l \mu_m AD}{(3-D)\beta b r^2} (l_{max})^{3-D} \left(\frac{1}{Y_{eff}} \frac{d\sigma}{dt}\right) \quad (5)$$

The only amounts that are explicitly time-dependent are $B$ and $\frac{d\sigma}{dt}$ so that at the end, the temporal evolution of stress is proportional to the temporal integral of the magnetic field:

$$\sigma(t) = k(Y_{eff}) \int B(t)dt \quad (6)$$

With $k(Y_{eff}) = \left( \frac{\sqrt{2}q_l \mu_m AD}{(3-D)\beta b r^2} \frac{(l_{max})^{3-D}}{Y_{eff}} \right)^{-1}$, $k$ in units of A/m/s, or magnetization per seconds. The Equation 6 shows that it is possible to use the magnetic field to measure the evolution of stress in laboratory rocks. While $k$ represents the geometric and mechanical properties of the source of electrification in laboratory rocks. If these experiments are correct, it would be expected that the magnetic field could reveal changes of stress on a geodynamic scale.

**3 b-value, earthquake entropy, magnetic field and critical system**

The seismicity of an area is statistically determined by Gutenberg-Richter's law on a geodynamic scale (Gutenberg and Richter, 1944). This law shows the number of earthquakes $N$ with magnitude equal to or greater than $M$ under the logarithmic relation: $\log N = a - bM$ and where parameters $a$ and $b$ depend on each study area. Each earthquake is generated by a sudden release of energy that is not recovered, so the Gutenberg-Richter's law describes the occurrence of a set of irreversible events (e.g. Stein and Wysession, 2003). Since parameters $a$ and $b$ give information about the stress conditions in which these irreversible events occur, De Santis et al. (2011) developed the concept of earthquake entropy $H$ based on Shannon entropy. Shannon's entropy measures the information of a system and its changes, however, the information of this system corresponds to the stress states of the lithosphere. In this way, the concept of earthquake entropy can be understood as the measure of the transition between different states of stress in the lithosphere. Using this, De Santis et al. (2011) found that the temporal variation of b-value of Gutenberg-Richter's law is related to earthquake entropy H(t) through:

$$b(t) = b_{max} 10^{-H(t)} \quad (7)$$

Where $b_{max} = e \log_{10} e$, which is constant. As H(t) can be understood as the measure of lithospheric stress (De Santis et al., 2011), the earthquake entropy can be directly related to stress through: $H(t) \equiv k_0 \sigma(t)$, where $k_0$ is in units of inverse stress. If the result shown by Equation 6 is self-similar and is also applicable at geodynamic scale, it implies that the b-value of Gutenberg-Richter's law (Equation 7) can be temporarily related to the magnetic field (Equation 6) by means of:

$$b(t) = b_{max} 10^{-k_0 k(Y_{eff}) \int B(t)dt} \quad (8)$$

On the other hand, De Santis et al. (2017) and Marchetti and Akhoondzadeh (2018) found that the daily accumulation of magnetic field anomalies before and after the Nepal 2015 Mw7.8 and Mexico 2018 Mw8.2 earthquakes had a behavior similar to that of a critical system so the shape of the magnetic field can be approximated to a sigmoid function: $B \sim \left( 1 + e^{-(t-t_0)} \right)^{-1}$ (Figure 4 upper panel). The integral of the sigmoid is shaped: $\ln(1 + e^{t-t_0})$, so by choosing $k(Y_{eff}) = 1$ and $t_0 = 10$ in Equation 8, it may show the b-value temporal evolution (Figure 4 lower panel). In it, the b-value decreases before an earthquake, suggesting that there must be a change in the lithospheric regime (to an imminent collapse) because of increased seismicity prior to the occurrence of an earthquake, i.e., the existence of seismic or foreshock swarms (Schorlemmer et al., 2005; Ruiz and Madariaga, 2018). This is consistent with other research that suggests that a b-value decrease may serve as an earthquake predictor since a decreasing b-value means that earthquakes of higher magnitudes are required in order to satisfy the Gutenberg-Richter's law (Imoto, 1991; Kulhanek et al., 2018).

**4 Seismic Moment, Moment Magnitude and Co-Seismic Magnetic Field**

The area $S$ that is implicit in the factor $A$ in Equation 4 is considered to calculate the co-seismic magnetic relation $B_{cs}$ with earthquakes, since it may correspond to the rupture area (Turcotte, 1997):

$$S \approx \frac{B_{cs}r^2}{\mu_m J_2} \frac{(3-D)}{D(D-2)} \left(l_{min}^{2-D}\right)\left(l_{max}^{D-3}\right) \quad (9)$$

By replacing Equation 9 in the Scalar Seismic Moment equation $M_0$ ($M_0 = \mu A d \approx \mu S d$), where $\mu$ is the shear modulus and $d$ the average slip) there is (Aki, 1966):

$$M_0 \approx \mu \frac{B_{cs}r^2}{\mu_m J_2} \frac{(3-D)}{D(D-2)} \left(l_{min}^{2-D}\right)\left(l_{max}^{D-3}\right)d \quad (10)$$

With the Scalar Seismic Moment it is possible to calculate the Moment Magnitude Scale $Mw$ ($Mw = \frac{2}{3}\log_{10}[M_0 \times 10^7] - 10.7$, for $M_0$ in $Nm$ units, and where $10^7$ has $(Nm)^{-1}$ units, Hanks and Kanamori, 1979). Then, according to the co-seismic magnetic field the Moment Magnitude is:

$$Mw \approx \frac{2}{3}\log_{10}\left[\left(\mu \frac{B_{cs}r^2}{\mu_m J_2} \frac{(3-D)}{D(D-2)} \left(l_{min}^{2-D}\right)\left(l_{max}^{D-3}\right)d\right) \times 10^7\right] - 10.7 \quad (11a)$$

If we consider the fractal dimension of granite ($D = 2.6$) (Turcotte, 1997) we have a more compact version of Equation 11a:

$$Mw \approx \frac{2}{3}\log_{10}\left[\left(\frac{1}{3.9} \frac{\mu}{\mu_m} \frac{B_{cs}}{J_2} \frac{dr^2}{(l_{min}^{0.6})(l_{max}^{0.4})}\right) \times 10^7\right] - 10.7 \quad (11b)$$

Utada el al., (2011) reported a variation of $B_{cs} = 0.8$ nT at a distance $r$ of the order of 100km from the fault plane during the Tohoku Earthquake 2011 Mw9.0 (Table 1). If we consider a minimum fracture of $l_{min} \approx 10^{-3}$ m (Shah, 2011), for granite $\mu_m = 13.5 \times 10^{-7}$ N/A$^2$ (Scott, 1983), $J_2 = 5 \times 10^{-6}$ A/m$^2$ (Tzanis and Vallianatos, 2002). In addition to the data provided by the USGS S = $625 \times 260$ km$^2$, $d = 5.27$ m, $\mu = 57$ GPa, where $l_{max} = \sqrt{S/\pi}$, the Moment Magnitude calculated with the magnetic field must be:

$$Mw \approx \frac{2}{3}\log_{10}[(4.1463 \times 10^{22}) \times 10^7] - 10.7 = 9.0 \quad (12a)$$

On the other hand, Johnston et al. (2006) reported changes in the magnetic field at several stations fairly close to Parkfield 2004 M6.0 earthquake (Table 1). For instance, the station GDM (Latitude: 35.8420; Longitude: -120.3380) measured a variation of $B_{cs} = 0.3$ nT at a distance $r \approx 2.5$ km from the fault. Using the general values $\mu_m, J_2$ and $l_{min}$ and the earthquake information: $\mu = 30$ GPa (Barbot et al., 2009), S $\approx 20 \times 10$ km$^2$ (Kim and Dreger, 2008) and $d = M_0/(\mu S) = 0.22$ m, with $M_0 = 1.3 \times 10^{18}$ Nm (Kim and Dreger, 2008). Moment Magnitude calculated with the magnetic field is:

$$Mw \approx \frac{2}{3}\log_{10}[(8.1545 \times 10^{17}) \times 10^7] - 10.7 = 5.9 \quad (12b)$$

The last example corresponds to the Loma Prieta 1989 M7.1 earthquake (Table 1). During the earthquake, at a distance of $r \approx 7$ km (Corralitos station) a peak of 0.9 nT that excelled the intense (non-seismic) magnetic noise was measured (Fenoglio et al., 1995; Karakeliana et al., 2002; Thomas et al., 2009). Using the same values of this section $\mu_m, J_2$ and $l_{min}$ and for this earthquake: $B_{cs} = 0.9$ nT, $r \approx 7$ km

(Karakeliana et al., 2002), μ = 30 GPa and S ≈ 40 × 10 km² (Wallace and Wallace, 1993), $d = 1.2$ (Berkeley Seismology Lab), the Moment Magnitude calculated is:

$$Mw \approx \frac{2}{3}\log_{10}[(9.1073 \times 10^{19}) \times 10^7] - 10.7 = 7.2 \quad (12c)$$

The result of Equation 12a, b and c are similar to the real one, therefore Equation 11 is valid for the following analyses. The expected co-seismic magnetic field can be obtained from Equation 11b in accordance with distance:

$$B_{cs} \approx 3.9\frac{\mu_m}{\mu}\frac{J_2\left(l_{min}^{0.6}\right)\left(l_{max}^{0.4}\right)}{d\,r^2}10^{\frac{3}{2}(Mw+6)} \quad (13)$$

The factor $10^{\frac{3}{2}(Mw+6)}$ here holds Nm units. Keeping the same values of $\mu_m$, $J_2$ and $l_{min}$ used so far, plus the data for the Tohoku 2011, Maule 2010, Sumatra 2004, Illapel 2015, Parkfield and Loma Prieta earthquakes (Table 1) the expected co-seismic magnetic variation for these events can be observed in Figure 5. This Figure also shows that co-seismic magnetic variations can reach hundreds of kilometers of radial distance from the rupture area. Even these variations can reach the ionosphere (48 km high from Earth's surface https://www.nasa.gov/mission_pages/sunearth/science/atmosphere-layers2.html), which could disturb the electron density within the ionosphere (Astafyeva et al., 2013; Kelley, 2017; Marchetti and Akhoondzadeh, 2018; Potirakis et al., 2018b). According to Kelley et al. (2017), it is possible to propagate a disturbance in the ionosphere if there is an electric field of the order of ~ 0.5 mili Volt/meter at ~ 90 km from the earth's surface. This is ~ $10^{-3}$ nT in magnetic terms if we consider $E = cB$, with $c = 3 \times 10^8$m/s, the speed of light. Kelley et al. (2017) also claim that the electrical disturbance required at Earth's surface should be close to ~ 0.2 V/m or ~ 0.5 nT. Figure 5 shows that the condition of ~ $10^{-3}$ nT at ~ 90 km from the earth's surface and ~ 0.5 nT at Earth's surface (~ $10 - 20$ km from epicenter) is reached for all earthquakes studied whit moment magnitude greater than ~Mw7. Therefore, ionospheric disturbances would not be expected for earthquakes with moment magnitudes less than ~Mw7.

**5 Ultra Low Frequency Magnetic Signals**

After establishing the magnitude of the expected co-seismic magnetic field, it is necessary to determine the order of magnitude of the oscillations present in the magnetic field. With this purpose, we consider that the current density is oscillating and can be expressed as a function of the polarization density as: $J = \dot{P} = \omega P_0$, so replacing the above in Equation 13 the following result is obtained:

$$\omega \approx \frac{1}{3.9}\frac{\mu}{\mu_m}\frac{dr^2 B_{cs}}{\left(l_{min}^{0.6}\right)\left(l_{max}^{0.4}\right)P_0}10^{-\frac{3}{2}(Mw+6)} \quad (14)$$

Where $P_0 = \delta\Lambda q_l dx/\sqrt{2}$ (Vallianatos and Tzanis, 1998), where the displacement of the fracture $dx$ is normally comparable to the Burgers vector and has a typical value of $5 \times 10^{-10}$ m (Slifkin, 1993), a minimum excess dislocation $\delta\Lambda = 1 \times 10^8$ m⁻² in semiconductor materials (JAMS-CS, 1999) and the electrical charge line $q_l \sim 10^{-11}$ C/m (Slifkin, 1993). Considering $l_{min} \approx 10^{-3}$ m (Shah, 2011), $\mu_m = 13.5 \times 10^{-7}$ N/A² (Scott, 1983), $J_2 = 5 \times 10^{-6}$ A/m² (Tzanis and Vallianatos, 2002). Also the data for the 2010 Tohoku earthquake from Table 1 and $B_{cs} = 0.8$ nT and $r = 100$ km (Utada el al., 2011), the frequency of the magnetic field oscillation associated with the 2011 Tohoku earthquake is of the order of $10^6$Hz; however, the co-seismic displacement $dx$ is not comparable to the Burgers vector but to the average displacement $d$, i.e., $dx \approx d = 5.27$ m, so the magnetic field oscillation frequency is:

$$\omega \sim 1.7 \, mHz \quad (15)$$

Oscillations of the order of mHz have been detected by De Santis et al., (2017), which is consistent with Equation 15, although frequencies of the order of µHz have been detected by Cordaro el at. (2018). However, according to Vallianatos and Tzanis (2003), the frequency of magnetic field oscillation associated with earthquakes is manifested in a range of at least three orders of magnitude, and this coincides with the measurements of Cordaro el at. (2018) (µHz) and De Santis et al., (2017) (mHz). The above information implies that in order to generate oscillation frequencies of the magnetic field in the pre-seismic stage similar to the co-seismic frequencies, polarizations $P_0$ and current densities $J$ within lithosphere should be similar to those found in the co-seismic stage ($P_0 \sim 3.7 \times 10^{-3}$ C/m² and $J \sim 5 \times 10^{-6}$ A/m²) and even these electrical conditions should be in some places of the lithosphere away from the fracture zone (main fault) (Scoville, et al., 2015). On the other hand, if the polarization is similar and the current density is lower, frequencies lower than those presented in Equation 15 are obtained. For example, if the lithosphere polarization is maintained in the pre-earthquake stage and the current density decreases by two orders of magnitude (i.e. $J \sim 10^{-8}$ A/m²) it is possible to obtain frequencies of the order of the µHz ($\omega = J/P_0 \sim 10^{-6}$ Hz), which means that according to Equation 1, to create lower magnetic frequencies there must be a lower stress change.

On the other hand, equation 14 depends on $l_{max}$ and corresponds to the maximum radius of the rupture area of an earthquake. This implies that at other times there will be a lower $l_{max}$ and therefore higher frequencies. In addition, we must remember that $l_{max}$ was calculated using the microcracks fractality. This means that $l_{max}$ can have a large range of orders of magnitude. Therefore, the oscillation frequency of the magnetic field associated with earthquakes must also have a fractal nature. This fractal property in magnetic measurements had already been found by other researchers prior to the occurrence of earthquakes (e. g. Potirakis et al., 2017 and references therein).

**6 Location of microcracks**

Kelley et al. (2017) show that it is necessary to have close to 0.5 nT at earth's surface in order to propagate a disturb in the ionosphere. If we consider that Marchetti and Akhoondzadeh (2018) found anomalous behaviors in the magnetic field using satellites, it can be suggested that $\sim 0.1 - 0.5$ nT is the magnetic variation created in the lithosphere prior to the occurrence of an earthquake. However, it is necessary to estimate the place in the lithosphere where these cracks might be occurring. It is also necessary to determine the order of magnitude of the microcracks dimensions within lithosphere.

Cordaro et al., (2018) show disturbances in magnetic field prior 2010 Maule earthquake (36∘17'24.0"S 73∘14'20.4"W). If we consider the OSO station (40∘20'24"S, 73∘05'24.0"W), we can note that it is $\sim$ 450 km from the epicenter of the 2010 Maule earthquake (the closest magnetic station to earthquake). As in this case we only want to calculate the orders of magnitude of the microcracks and their location, we will consider the general version of Equation 2, which is shown in Equation 16 (Griffiths, 1996; Vallianatos and Tazanis, 2003).

$$\vec{B}(x_0, y_0, h) = \frac{3\mu_m V}{8\pi} \frac{\vec{J} \times \vec{r}}{r^3} \quad (16)$$

Where $V$ is the fractal volume defined in Equation 3, $x_0$ and $y_0$ are the point near the surface of the lithosphere where the station is located, and $h$ the depth of the microcrack. This depth $h$ corresponds to the semi brittle-ductile transition and is between 10 and 20 km deep (Scholz, 2001; Sun, 2011). For these

calculations we will consider that $h = -15$ km. If we consider that the microcracks are occurring in the future earthquake rupture zone, in addition to the data in Table 2, it would imply that the microcracks would have dimensions of the order of $\sim 300$ m to obtain more than $\sim 0.2$ nT at $\sim 450$ km. The result of using this microcrack length and Table 2 is shown in Figure 6. Using the same values, we find that greater magnetic variations exist closer to the future seismic rupture zone. For example, within a radius of 100 km there are magnetic variations of 10 nT (white circle in Figure 7), while within a radius of 10 km there would be variations of the order of 160 nT (magenta circle in Figure 7). These variations have never been recorded, therefore microcracks cannot be of the order of hundreds of meters, but must be smaller. Neither can they come from the future seismic source.

On the other hand, if we consider that microcracks are occurring near the stations, it is enough to take an $\sim 30$ m to obtain magnetic variations similar to those suggested Kelley et al. (2017) at Earth's surface. Figure 8 it is shown that with this configuration the measurements can be replicated. However, it is necessary that microcracks with $l_{max}$ of the order of tens of meters should be occurring in different places of the lithosphere.

**7 Summary and conclusions**

This work studied the role of the magnetic field in the lithospheric dynamics; specifically, the physics that could be associated with various measurements that relate magnetic fields and earthquakes in a complete cycle, i.e. from a stress disturbance to the magnetic frequencies correlated with the occurrence of an earthquake. The results of each section are below:

Since a change in stress could trigger an earthquake, section two discussed the way a change in stress causes fractures within the rocks, the flow of electrical currents and the generation of magnetic fields. Therefore, the goal of this section was to achieve a relationship (equation 6) between the temporal evolution of stress with the integral over time of the magnetic field through a constant $k$. It was also possible to store all the electrical and mechanical information of the rocks in the constant $k$, which represent the magnetization per second of the rocks.

The goal of section three is of great relevance since it established a relationship between the behavior of the magnetic field (critical system) and a b-value decrease of the Gutenberg-Richter Law before and after the occurrence of earthquakes through earthquake entropy concept (Equation 8 and Figure 4). This was possible by assuming that the behavior of laboratory samples would exhibit the same physics as lithospheric rocks. Another goal of this section was to obtain a more physical interpretation about the entropy of earthquakes, its relation with magnetism and the impending earthquakes: As entropy can be considered as the energy diffusion of a system, the accumulation of stress (energy) in the lithosphere (open system) must be diffused. This means that the increment in the number of magnetic anomalies and their relationship with an increase in seismicity (earthquake swarms and/or seismic precursors) prior to the occurrence of large earthquakes are part of the energy diffusion mechanisms. However, this may also be interpreted inversely: The non-existence of seismic and magnetic precursors could violate the second law of thermodynamics. However, more studies are needed to corroborate whether the emission of magnetic signals really has any relationship with the entropy of earthquakes.

The great goal of section 4 was to find and corroborate an analytical relationship between co-seismic magnetic measurements and the magnitude of earthquakes (Equations 11a, b). It was possible to obtain Equations 11a, b by considering the area of rupture of the earthquake as a crack of the MCD model.

Another goal of this section was to find an analytical relationship that would allow to determine the magnitude of co-seismic magnetic signals as a function of the epicentral distance (Equation 13). Figure 5 shows the intensity of the expected co-seismic magnetic variation for several earthquakes as a function of the distance to the area of rupture. It is observed that magnetic variations can easily reach the ionosphere for earthquakes of magnitudes greater than Mw8.3 (dashed blue line). Many magnetometers have the resolution of 0.1 nT (dashed red line) so magnetic variations produced by large earthquakes (~Mw9) could be detectable by magnetometers several hundred kilometers from the area of rupture. However, it is not expected that the magnetometers can detect magnetic variations related to small earthquakes, i.e. magnitudes much lower than Mw8.0 and tens of km from the source. For instance, during the L'Aquila 2009 M6.1 earthquake (central Italy), large magnetic variations were reported associated with displacements of the instruments due seismic waves (0.8 nT) at 6.7 km away from the source of the earthquake (Nenovski, 2015; Masci and Thomas, 2016). However, using $r \approx 6.7$ km (Nenovski, 2015), $Mw = 6.1$, $\mu = 32 \times 10^9$ Pa , $d = 0.4$ m and $S = 19 \times 13$ km$^2$ (Walters et al., 2019) and the same values of $\mu_m$, $J_2$ y $l_{min}$ used in section 5 into Equation 13 the expected co-seismic magnetic field is $B_{cs} \approx 0.04$ nT, which is quite close to the instrumental noise of the L'Aquila station (0.02 nT) (Villante et al., 2010), making these magnetic co-seismic variations almost undetectable.

The goal of section 5 was to theoretically find the oscillation frequencies of the magnetic field that may be related to the occurrence of earthquakes. They were found to have frequencies of the order of mHz. The existence of frequencies of different orders of magnitude or fractal nature of oscillations prior to earthquakes were also analyzed. It is concluded that for there to be magnetic variations in the lithosphere prior to earthquakes it is necessary that the conditions of polarization and density of currents are similar to those that can be found in the co-seismic stage. All these magnetic variations are part of the ULF reported by several authors.

Section 6 looked for the location of the microcracks and their size. It was found that microcracks are unlikely to be created in the future seismic rupture zone. However, if microcracks of the order of 30 m exist at depths of 10-20 km, it is possible to explain the expected magnetic variations ($\sim 0.2$ nT). This implies that microcracks must be occurring throughout the lithosphere due to a change in the stress field.

On the other hand, the physics of the co-seismic stage (Section 4) and the stage prior to earthquakes (section 6) could be the same: microcracks. Where the only difference comes from the size of $l_{max}$. This is relevant since in the future it will be necessary to investigate microcracks as a factor that allows propagation of seismic fractures. In addition, it will also be necessary to study the distribution of microcracks throughout the lithosphere. This would allow estimating the places where it is more likely to find magnetic variations as well as possible future earthquakes.

Finally, it can be concluded that the controversial magnetic phenomena registered by different research groups, behavior of cumulative daily number of magnetic anomalies, co-seismic magnetic field and oscillation frequencies of the magnetic field, can all have the same and unique physical origin: the cracking of brittles and semi brittles materials of the crust due to stress changes. However, there is still no clarity about how these stress changes can generate the nucleation of earthquakes. Therefore, future studies should focus on interpreting magnetic records as a tool to measure stress changes in the lithosphere. Especially when there are no appreciable deformations of the lithosphere. This could provide new information to seismic source studies.

**Acknowledgements**

P.V.-A. acknowledges Patricia Aravena, Alejandro Venegas, Patricia Venegas and Richard Sandoval for outstanding support to carry out this work, and Valeria Becerra-Carreño for her scientific support. D. L. acknowledges partial financial support from Centers of excellence with BASAL/CONICYT financing, grant FB0807, CEDENNA.

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

**Caption Tables**
Table 1: Earthquake data from Tohoku 2009 (USGS), Maule 2010 (Vigny et al., 2011; Yue et al., 2014),
Sumatra 2004 (Menke et al., 2006), Illapel 2015 (Tilmann et al., 2016; Shrivastava et al., 2016), Parkfield
2004 (Kim and Dreger, 2008; Barbot et al., 2009) and Loma Prieta 1989 (Berkeley Seismology Lab;
Wallace and Wallace, 1993).
Table 2: Typical values and inputs to Equations 3 and 16.
**Captions Figure**
Figure 1: Schematic description of the generation of microcracks and currents due to mechanical stresses
on rocks. a)  A moving edge dislocation meets a barrier or obstacle. b) A set of edges dislocations are
piled up generating a microcracks (blue triangle). The microcracks generate the breaking of ionic bonds,
which allows polarization of the microcracks. c) Microcracks can propagate through different paths (blue
lines). d) An avalanche of microcracks can cause larger scale cracks.
Figure 2: Outline of the experiments carried out with rocks during compressive modes. a) The change of
effort $\sigma$ generates one failure of the rock at an angle $2\theta$. The black arrows indicate the relative slip within
the rock. b) Electrification of the rock in microcracks zones close to the fault. The yellow arrows indicate
the direction of the generated currents.
Figure 3: Schematic magnetic field measured in an interface due to a polarized sphere of volume V
embedded in a medium with magnetic permeability $\mu_m$.
Figure 4: Upper: Temporal evolution of the magnetic field in the form of a critical system (De Santis et
al., 2017, Marchetti and Akhoondzadeh, 2018). Lower: Temporal evolution of b-value prior to an
earthquake. The vertical line indicates when an earthquake occurs according to De Santis et al. (2017).
Figure 5: Expected co-seismic magnetic field as a function of distance for the Tohoku 2011, Maule 2010,
Sumatra 2004, Illapel 2015 earthquakes and Parkfield 2004 (see Table 1 for earthquakes information).
Figure 6: Total Magnetic Field Intensity at the Earth's surface using parameters of Table 2 and $l_{max} \approx$
300 m in Equation 3 and 16. The domain is $[-1000, 1000] \times [-1000, 1000] \times [-20, 0] \ km^3$. Values
greater than 0.2 nT can be observed in OSO station (close to 450 km from future Maule earthquake). The
red star show the hypocenter of the future earthquake and the yellow arrow is the direction of the electric
current $J$.

Figure 7: Total Magnetic Field Intensity at the Earth's surface using the same parameters of Figure 6. However, in this Figure we indicate the places where is possible found magnetic variations of 10 nT (white circle) and 160 nT (magenta circle). These variations have never been recorded.

Figure 8: Total Magnetic Field Intensity at the Earth's surface using parameters of Table 2 and $l_{max} \approx$ 30 m in Equation 3 and 16. The domain is $[-200, 200] \times [-200, 200] \times [-20, 0]\ km^3$. Values greater than 0.2 nT can be observed in OSO station. The yellow arrow is the direction of the electric current $J$. This size of microcracks could be the one that allows to explain the measurements of magnetic variations of Cordaro et al. (2018), Marchetti and Akhoondzadeh (2018) and the suggestion of Kelley et al. (2017).

**Table 1**

|  | Tohoku Mw9.0 (Japan) | Maule Mw8.8 (Chile) | Sumatra Mw9.3 (Indonesia) | Illapel Mw8.3 (Chile) | Parkfield Mw6.0 (California, USA) | Loma Prieta Mw7.1 (California, USA) |
|---|---|---|---|---|---|---|
| Latitude Longitude | 38.322 142.369 | -36.290 -73.239 | 3.316 95.854 | -31.573 -71.674 | 35.815 -120.374 | 37.040 -121.877 |
| $\mu$ [Pa] | $5.7 \times 10^{10}$ | $3.3 \times 10^{10}$ | $7 \times 10^{10}$ | $3.5 \times 10^{10}$ | $3 \times 10^{10}$ | $3 \times 10^{10}$ |
| d [m] | 5.27 | 4 | 5 | 5 | 0.22 | 1.2 |
| S [km²] | $625 \times 260$ | $450 \times 120$ | $1200 \times 200$ | $200 \times 80$ | $20 \times 10$ | $40 \times 10$ |

**Table 2**

| parameter | Value | reference |
|---|---|---|
| $\mu_m$ (Granite) | $13.5 \times 10^{-7}\ N/A^2$ | Scott, 1983 |
| $J$ | $5 \times 10^{-6}\ A/m^2$ | Tzanis and Vallianatos, 2002 |
| $D$ (Granite) | 2.6 | Turcotte, 1997 |
| $\theta$ (Granite) | 69.93° | Yin et al., 2018 |
| $l_{min}$ (Granite) | $10^{-3}$ m | Shah, 2011 |
| $l_{max1}$ | 300 m | Input |
| $l_{max2}$ | 30 m | Input |
| $h$ | 15 km | Input |

**Figure 1**

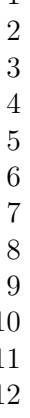

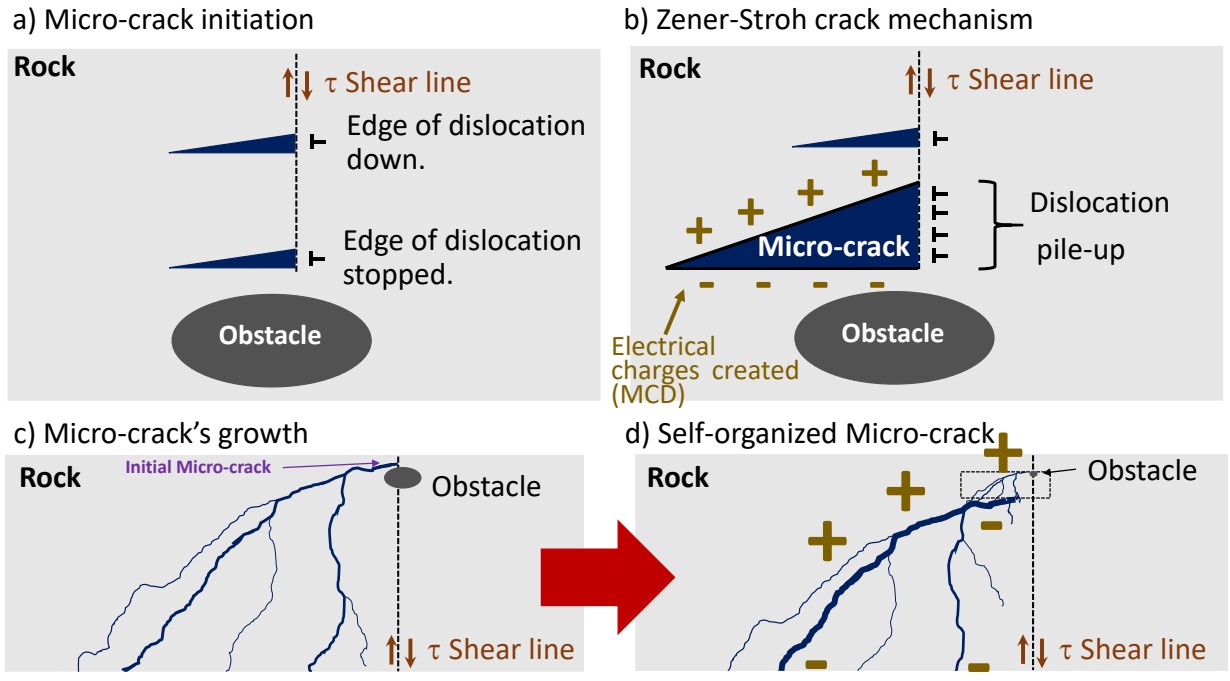

**Figure 2**

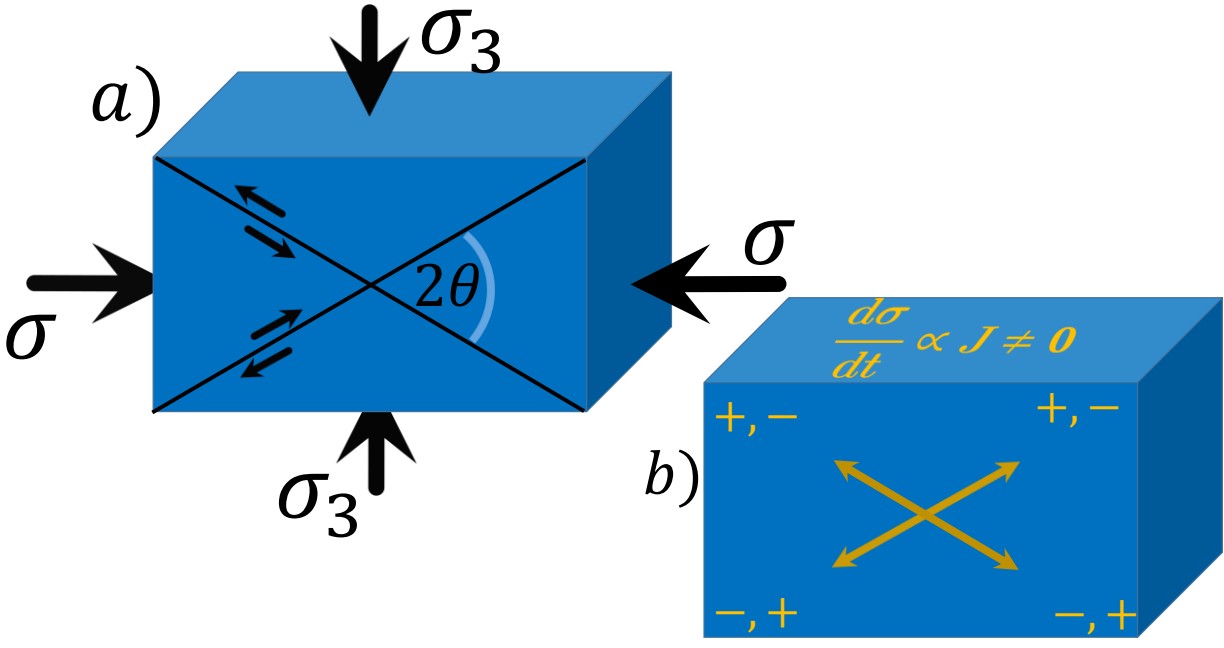

a)

$\sigma_3$

$\sigma$

$\sigma$

$2\theta$

$\sigma_3$

b)

$\frac{d\sigma}{dt} \propto J \neq 0$

$+, -$      $+, -$

$-, +$      $-, +$

**Figure 3**

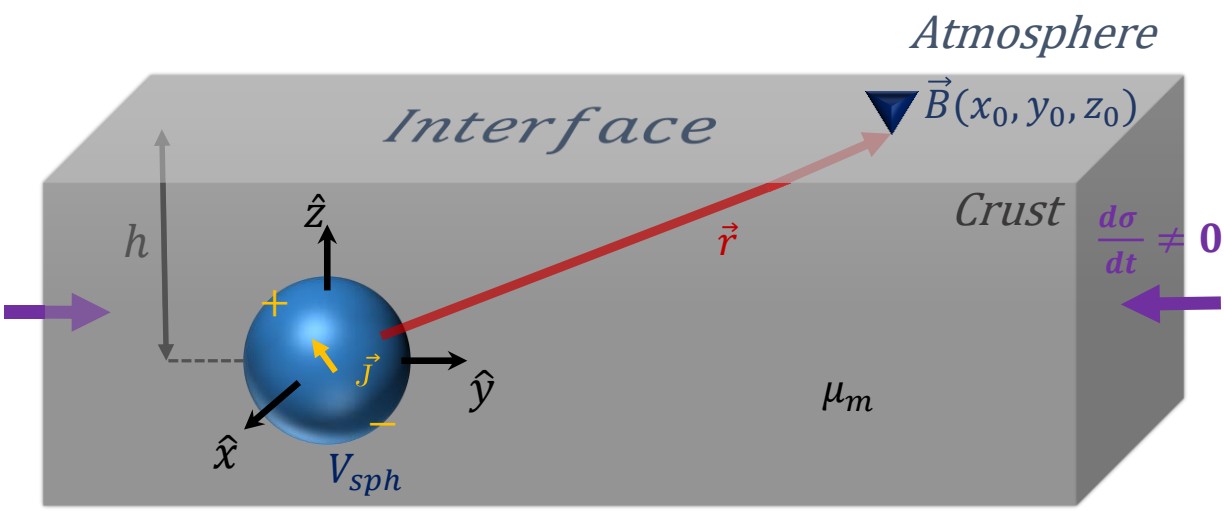

Atmosphere

Interface

$\vec{B}(x_0, y_0, z_0)$

Crust

$\frac{d\sigma}{dt} \neq 0$

$\hat{z}$

$h$

$+$

$\vec{r}$

$\vec{J}$

$\hat{y}$

$\mu_m$

$\hat{x}$

$-$

$V_{sph}$

**Figure 4**

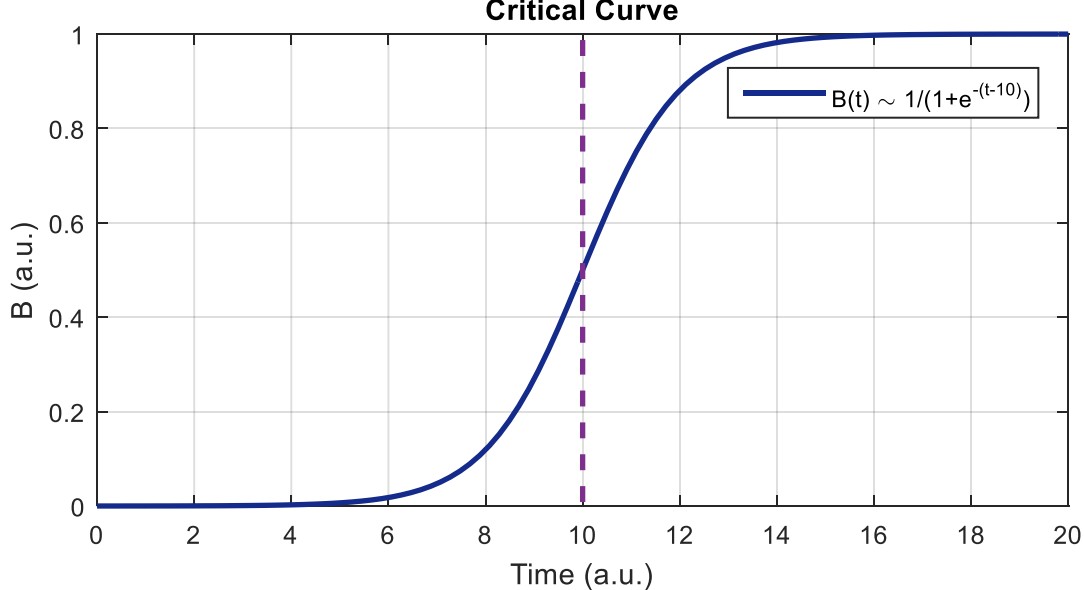

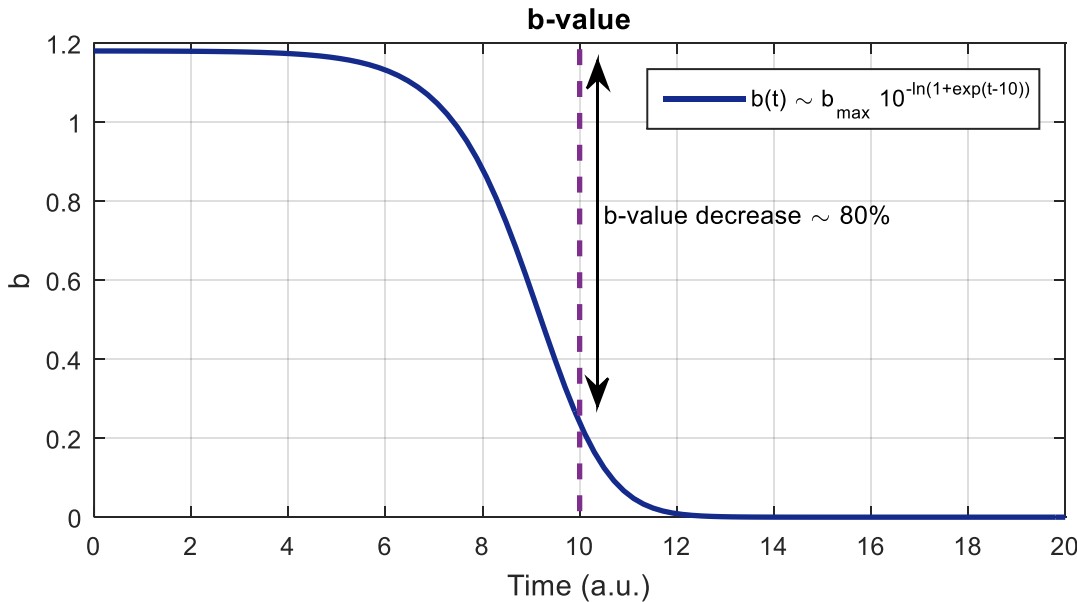

**Figure 5**

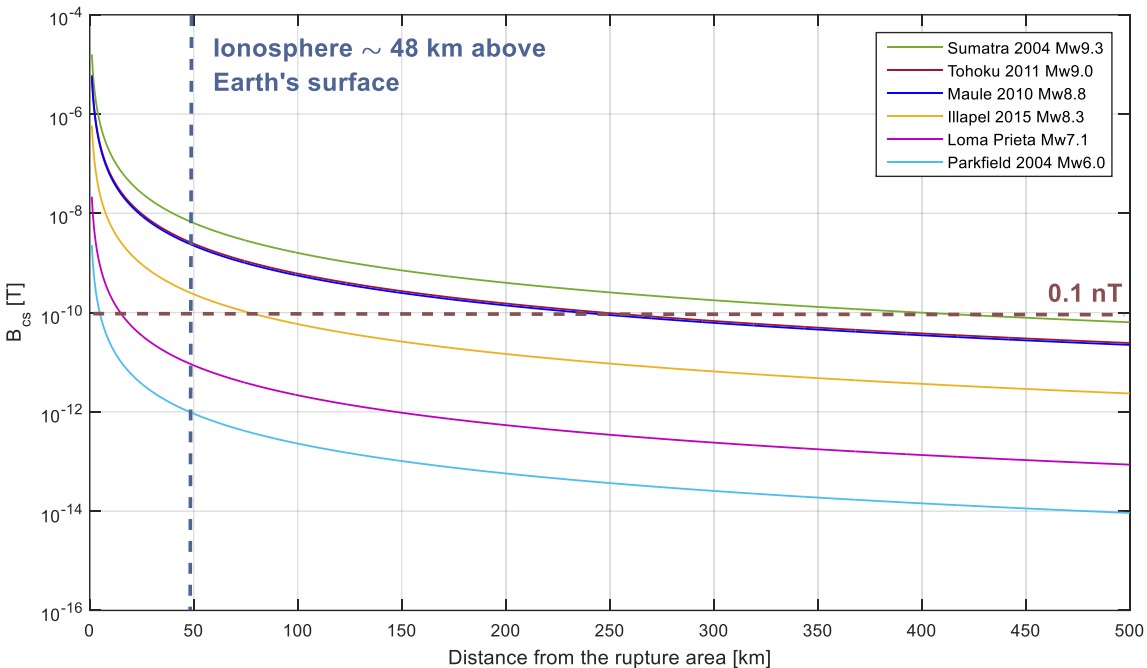

**Figure 6**

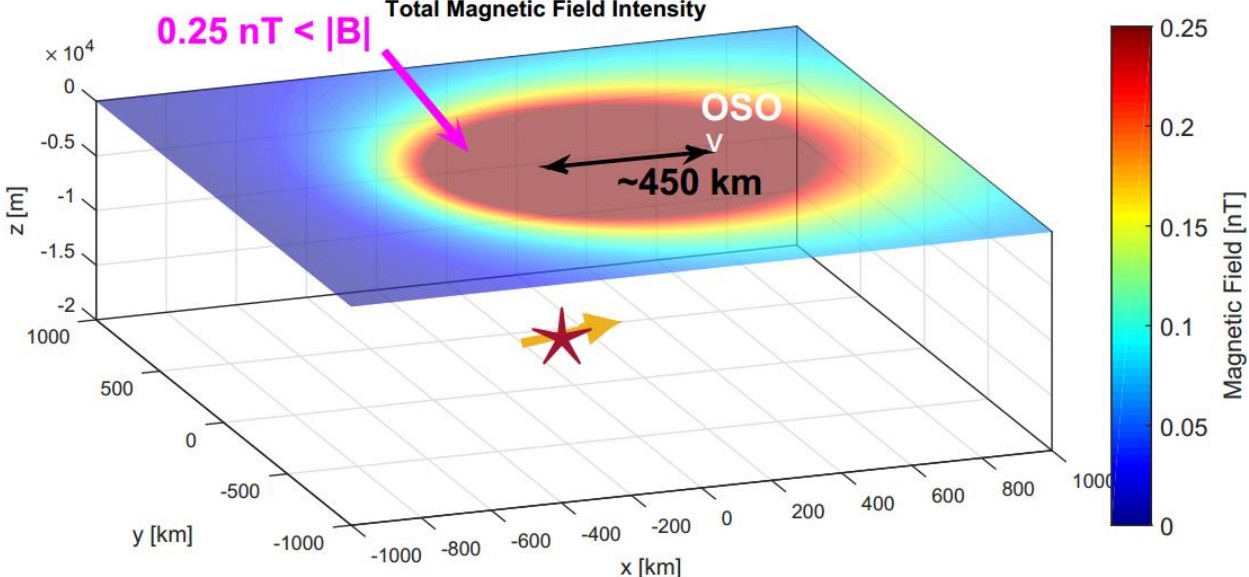

**Figure 7**

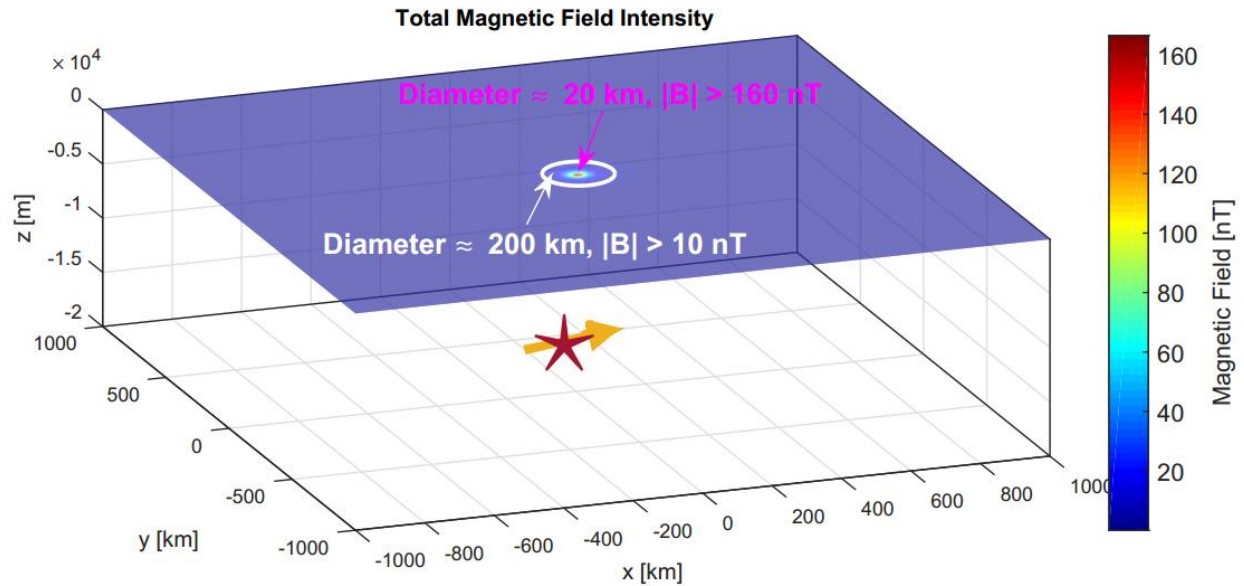

**Figure 8**

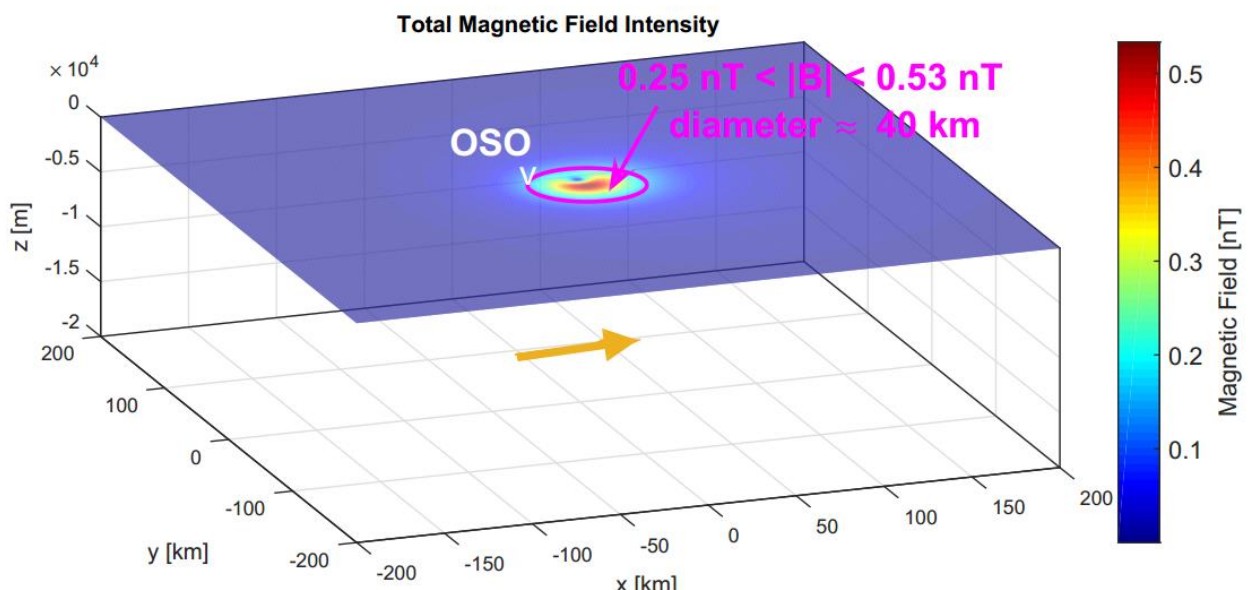