# Peer review of "A review and upgrade of the Lithospheric dynamics in context of the"

_Natural Hazards and Earth System Sciences, 2019_

## Referee Comment (RC1) · Michael E. Contadakis (Referee) · 20 Mar 2019

This work is a meticulous review with some ameliorations of the Lithospheric dynamics within the context of Seismo-electromagnetic Theory, with the scope to offer a possible physical casual connection between Lithospheric stress change and magnetic field variations for four empirically observational results which indicate relationship of magnetic field with earthquakes, Namely a) The Motion of charge edge dislocation model, b) The resulting electric current from stress variations c) The Seismic Moment and earthquake Moments and d) The distant dependent co- seismic magnetic field variations. The paper is very well written and his subject is very interesting to scientist

working in related field. For instance for me the fourth issues is of great interest. In concluding I suggest that the paper should be accepted for publication after some careful reading in order to correct some minor pitfalls, like the one on page 2,line5. . .study the any change. . . . study any change. . . .. And so on.

---

## Referee Comment (RC2) · Angelo De Santis (Referee) · 20 Mar 2019

This paper presents the theory underlying four different empirical observations showing a possible link between magnetic fields and earthquakes. In my opinion, the most significant contribution is the attempt to connect the theory of preseismic and coseismic magnetic effects under the same umbrella.

The paper is fairly written and organized. However, there is some confusion in presenting some equations and some parameters are not clearly defined.

The requested revision is a minor/intermediate-level revision. I suggest some changes

and corrections after which the work can be published.

Major points

1. It seems to me that there is some confusion in some equations and the corresponding definitions of some parameters. Below some details:

Pag.4, line 4. V is not explicitly defined. I suggest to write: ". . . to the sphere and V the volume . . ."

Pag.4, eq. (3) and line 14. SR is about 1, but it is not mentioned its meaning. Is it just a simple proportional factor? Does it depend on the microcracks geometry or spatial distribution? Or whatelse?

Pag.4, line 15. S is introduced here as contributing to the expression of A, and not directly in one equation. On the contrary, later on the text (Pag.5 line 35) it is said: "area S of Equation 4". By the way, which is the meaning of A? The difference between S and A should be stated clearly.

2. Pag. 6, Equations 12a, 12b, 12c. The estimated magnitudes are given with too high resolution. I suspect that estimating the associated errors of each parameter involved in the equations, the magnitude values could be given with a lower accuracy. Some short discussion on the involved errors should be given, and, in turn, the magnitude values should be given with less numerical figures.

3. Pag. 7, line 1. Here it is said that the ionosphere is at an altitude of 48km. Probably, this value in the mentioned web link was given as the altitude where the ionosphere starts (although other sources pose the value at around 60 km. see e.g. Wikipedia), till around 1000 km. Actually the highest ionospheric electron density is at around 300 km of altitude. By the way, the given link is now missing at NASA website.

4. Pag. 7, section 5. I was surprised about the large range of the magnetic field oscillation frequency from mHz to MHz. I think that some words would be necessary about this point.

Minor points

1. Pag.2, line 5. Please remove "the" before "any change".

2. Pag.3, line 7. Please write "explains".

3. Pag.3, line 25. Please correct as "MCD".

4. Pag.3, line 28. Please correct as "compressional".

5. Pag.4, line 20. Please write "corresponds".

6. Pag.4, line 25. Please change "y" with " and". (by the way, this mistake happens other times, e.g. pag.6, line 38, so I can presume that this text was originally translated from a Spanish text; for your safety, please check across the text if this mistake appears other times).

7. Pag.5, line 12. Please correct "untis" as "units".

8. Pag. 10, line14. Please remove a comma before "https. . ."

9. Pag.11, line 17. I think the Bibcode is not necessary (also because it is not fully written).

10. Pag. 11, line 34. Please correct "Fraser-Smitha" with "Fraser-Smith".

11. Pag.14, line 37. Please correct as "Vallianatos".

12. Pag.15, line 5. Please insert a blank between "Maule" and "Megathrust".

---

## Short Comment (SC1) · 8 Apr 2019

Dear referee M.C. Contadakis,

Thank you so much for your words. We will gladly improve those little mistakes in the next revision steps.

Best regards Patricio Venegas-Aravena on behalf of the authors
* * *

---

## Short Comment (SC2) · 8 Apr 2019

Dear referee A. De Santis,

Thank you very much for your comments and suggestions. We have read each of them carefully.

1.- - We will explain the volume V in the paragraph. - SR is a factor defined by ( $1 - (l\_min/l\_max)^{(3-D)}$ ) where $l\_min$ and $l\_max$ are at the length of the smallest and largest microcracks respectively. D is the fractal dimension. We assume that the ratio between large and small microcracks is zero case, which implies that SR is

approximately 1. If you think it is needed, we can add an appendix explaining the fractal volume. - A is a constant that appears to integrate fractal surface of microcracks. That is why it depends on an area S. - Area S is implicit in constant A. We improve the wording of those paragraphs on page 4 and 5.

2.- Unfortunately, the errors of many other authors parameters were not made explicit in their investigations. However, you are right in the amount of figures, we must reduce them.

3.- Excuse me, apparently the page changed link. This is the correct one now: https://www.nasa.gov/mission_pages/sunearth/science/atmosphere-layers2.html. The issue of the exact altitude of the ionosphere is not a problem for us. The latest research (e.g. 10.1002/2016JA023601) shows that it is possible to propagate a disturbance in the ionosphere ($\sim$0.5 mili Volt/meter or at 100 km above earth's surface) from the earth's surface ($\sim$0.2 V/m). In magnetic terms (assuming $E = cB$ with $E$, $B$ and $c$ electric, magnetic field and light velocity), a disturbance of the order of $\sim$0.5 nT is required at eath's surface to reach this condition. As shown in Figure 5, all earthquakes have amplitudes greater than 0.2 nT to about 10 km of the rupture, which also means that it is the magnetic amplitude at the earth's surface (or close to it). Using this model and magnetic magnitude order, it is also possible explain the magnetic perturbations shown in Nepal 2015 and Mexico 2017 as lithospheric origin (in principle).

4.- This point is interesting because equation 14 shows a dependence on larger microcracks $l\_max$ and polarization $P\_0$ of the lithosphere (more clear in text as $w = J/P\_0$). The rest is constant at a given distance. If $P\_0$ is also constant in the lithosphere, the frequency emitted in a zone will only depend on the larger microcracks created by stress changes. And since the microcracks will always be smaller than the earthquake size itself, added to the fractal nature of the cracks, it is natural that the frequency also takes several orders of magnitude greater. Even higher frequencies can be reached if the polarization is lower. Exactly this analysis also allows to explain some results of fractality in the magnetic frequencies found by other researchers (e.g.

10.1007/s11069-016-2558-8). That is why equation 14 represents a lower limit for a given constant P_0 inside the lithosphere.

The rest are minor modifications that we will correct when the journal allows us to edit our manuscript.

Thank you very much for your comments and suggestions. We believe that it has helped us to improve our manuscript. Best regards Patricio Venegas-Aravena on behalf of the authors

---

## Author Comment (AC1) · 5 May 2019

Dear referee M.C. Contadakis, We have already improved our manuscript. En el pdf en la respuesta a R2 se encuentra un pdf con las mejoras realizadas.

Best regards Patricio Venegas-Aravena on behalf of the authors.

---

## Author Comment (AC2) · 5 May 2019

Dear Prof. De Santis. We have already considered your thoughts and suggestions, the improvements made are in the pdf highlighted in green. In it the requested improvements are shown and a section was added (section 6) estimating the origin and size of the microcracks in the stage prior to earthquakes.

---

## Author Comment (AC3) · 5 May 2019

Dear Editor and referees, In this answer you will find the improvements made in the manuscript. They are highlighted in green. We have added section 6 where we explore in a simple way the place of the lithosphere where microcracks could occur. In addition we added three figures explaining the above.

We think that the work has improved substantially so we thank you. If you find errors, do not hesitate to contact us.

Best regards Patricio Venegas-Aravena on behalf of the authors.

[Figure]

Please also note the supplement to this comment:
https://www.nat-hazards-earth-syst-sci-discuss.net/nhess-2019-22/nhess-2019-22-AC3-supplement.pdf

―――――――――――――――――――――

[Figure]

**Supplement:**

[revised manuscript text omitted]
 $\sim 0.2$ V/m or $\sim 0.6$ nT. Figure 5 shows that the condition of $\sim 10^{-3}$ nT at $\sim 90$ km from the earth's surface and $\sim 0.6$ nT at Earth's surface ($\sim 10 - 20$ km from epicenter) is reached for all earthquakes studied whit moment magnitude greater than $\sim$Mw7. Therefore, ionospheric disturbances would not be expected for earthquakes with moment magnitudes less than $\sim$Mw7.

**5 Ultra Low Frequency Magnetic Signals**

After establishing the magnitude of the expected co-seismic magnetic field, it is necessary to determine the order of magnitude of the oscillations present in the magnetic field. With this purpose, we consider that the current density is oscillating and can be expressed as a function of the polarization density as: $J = \dot{P} = \omega P_0$, so replacing the above in Equation 13 the following result is obtained:

$$\omega \approx \frac{1}{3.9} \frac{\mu}{\mu_m} \frac{d r^2 B_{cs}}{(l_{min}^{0.6})(l_{max}^{0.4}) P_0} 10^{-\frac{3}{2}(Mw+6)} \quad (14)$$

Where $P_0 = \delta \Lambda q_l dx / \sqrt{2}$ (Vallianatos and Tzanis, 1998), where the displacement of the fracture $dx$ is normally comparable to the Burgers vector and has a typical value of $5 \times 10^{-10}$ m (Slifkin, 1993), a minimum excess dislocation $\delta \Lambda = 1 \times 10^8$ m$^{-2}$ in semiconductor materials (JAMS-CS, 1999) and the electrical charge line $q_l \sim 10^{-11}$ C/m (Slifkin, 1993). Considering $l_{min} \approx 10^{-3}$ m (Shah, 2011), $\mu_m = 13.5 \times 10^{-7}$ N/A$^2$ (Scott, 1983), $J_2 = 5 \times 10^{-6}$ A/m$^2$ (Tzanis and Vallianatos, 2002). Also the data for the 2010 Tohoku earthquake from Table 1 and $B_{cs} = 0.8$ nT and $r = 100$ km (Utada el al., 2011), the frequency of the magnetic field oscillation associated with the 2011 Tohoku earthquake is of the order of $10^6$Hz; however, the co-seismic displacement $dx$ is not comparable to the Burgers vector but to the average displacement $d$, i.e., $dx \approx d = 5.27$ m, so the magnetic field oscillation frequency is:

$$\omega \sim 1.7 \, mHz \quad (15)$$

Oscillations of the order of mHz have been detected by De Santis et al., (2017), which is consistent with Equation 15, although frequencies of the order of μHz have been detected by Cordaro el at. (2018).

However, according to Vallianatos and Tzanis (2003), the frequency of magnetic field oscillation associated with earthquakes is manifested in a range of at least three orders of magnitude, and this coincides with the measurements of Cordaro el at. (2018) (µHz) and De Santis et al., (2017) (mHz). The above information implies that in order to generate oscillation frequencies of the magnetic field in the pre-seismic stage similar to the co-seismic frequencies, polarizations $P_0$ and current densities $J$ within lithosphere should be similar to those found in the co-seismic stage ($P_0 \sim 3.7 \times 10^{-3}$ C/m$^2$ and $J \sim 5 \times 10^{-6}$ A/m$^2$) and even these electrical conditions should be in some places of the lithosphere away from the fracture zone (main fault) (Scoville, et al., 2015). On the other hand, if the polarization is similar and the current density is lower, frequencies lower than those presented in Equation 15 are obtained. For example, if the lithosphere polarization is maintained in the pre-earthquake stage and the current density decreases by two orders of magnitude (i.e. $J \sim 10^{-8}$ A/m$^2$) it is possible to obtain frequencies of the order of the µHz ($\omega = J/P_0 \sim 10^{-6}$ Hz), which means that according to Equation 1, to create lower magnetic frequencies there must be a lower stress change.

On the other hand, equation 14 depends on $l_{max}$ and corresponds to the maximum radius of the rupture area of an earthquake. This implies that at other times there will be a lower $l_{max}$ and therefore higher frequencies. In addition, we must remember that $l_{max}$ was calculated using the microcracks fractality. This means that $l_{max}$ can have a large range of orders of magnitude. Therefore, the oscillation frequency of the magnetic field associated with earthquakes must also have a fractal nature. This fractal property in magnetic measurements had already been found by other researchers prior to the occurrence of earthquakes (e. g. Potirakis et al., 2017 and references therein).

**6 Location of microcracks**

Cordaro et al. (2019b) showed that the intensity of the magnetic field that they considered anomalous prior to the occurrence of the earthquakes of Maule 2010, Iquique 2014 and Illapel 2015 were of the order of $\sim$ 0.2 nT. This value is close to the one that Kelley et al. (2017) indicates to propagate disturbances in the ionosphere. If we also consider that Marchetti and Akhoondzadeh (2018) found anomalous behaviors similar to those found by Cordaro et al. (2019b) in the magnetic field, but using satellites, it can be suggested that $\sim$ 0.2 nT is the magnetic variation created in the lithosphere prior to the occurrence of an earthquake. However, it is necessary to estimate the place in the lithosphere where these cracks might be occurring. It is also necessary to determine the order of magnitude of the microcracks dimensions within lithosphere.

If we consider the case of the OSO station (40∘20'24"S, 73∘05'24.0"W) in Cordaro et al. (2019b), 
[revised manuscript text omitted]

[Figure]

[Figure]

**Figure 4**

[Figure]

[Figure]

**Figure 5**

[Figure]

**Figure 6**

[Figure]

[Figure]

[Figure]

---

## Author Response (AR1)

Dear referees A. De Santis and M. Contadakis,

Thank you very much for your comments and suggestions. We have read each of them carefully and we think that all the corrections already done are improved the manuscript. We also add new typos corrections and a new section related with the localization or origin of anomalies. This response is almost the same that it had been uploaded in the discussion.

On the other hand, we enlist the previous corrections as below. All the manuscript's changes are highlighted in green.

###################################################################################

1.-
- We will explain the volume V in the paragraph.
- SR is a factor defined by ( $1 - (l\_min/ l\_max)^{(3-D)}$ ) where $l\_min$ and $l\_max$ are at the length of the smallest and largest microcracks respectively. D is the fractal dimension. We assume that the ratio between large and small microcracks is zero case, which implies that SR is approximately 1. If you think it is needed, we can add an appendix explaining the fractal volume.
- A is a constant that appears to integrate fractal surface of microcracks. That is why it depends on an area S.
- Area S is implicit in constant A. We improve the wording of those paragraphs on page 4 and 5.

2.- Unfortunately, the errors of many other authors parameters were not made explicit in their investigations. However, you are right in the amount of figures, we must reduce them.

3.- Excuse me, apparently the page changed link. This is the correct one now: https://www.nasa.gov/mission_pages/sunearth/science/atmosphere-layers2.html.
The issue of the exact altitude of the ionosphere is not a problem for us. The latest research (e.g. 10.1002/2016JA023601) shows that it is possible to propagate a disturbance in the ionosphere (~0.5 mili Volt/meter or at 100 km above earth's surface) from the earth's surface (~0.2 V/m). In magnetic terms (assuming E = cB with E , B and c electric, magnetic field and light velocity), a disturbance of the order of ~0.5 nT is required at eath's surface to reach this condition. As shown in Figure 5, all earthquakes have amplitudes greater than 0.2 nT to about 10 km of the rupture, which also means that it is the magnetic amplitude at the earth's surface (or close to it). Using this model and magnetic magnitude order, it is also possible explain the magnetic perturbations shown in Nepal 2015 and Mexico 2017 as lithospheric origin (in principle).

4.- This point is interesting because equation 14 shows a dependence on larger microcracks $l\_max$ and polarization $P\_0$ of the lithosphere (more clear in text as w = J/P\_0). The rest is constant at a given distance. If $P\_0$ is also constant in the lithosphere, the frequency emitted in a zone will only depend on the larger microcracks created by stress changes. And since the microcracks will always be smaller than the earthquake size itself, added to the fractal nature of the cracks, it is natural that the frequency also takes several orders of magnitude greater. Even higher frequencies can be reached if the polarization is lower. Exactly this analysis also allows to explain some results of fractality in the magnetic frequencies found by

other researchers (e.g. 10.1007/s11069-016-2558-8). That is why equation 14 represents a lower limit for a given constant $P\_0$ inside the lithosphere.

The rest are minor modifications that we will correct when the journal allows us to edit our manuscript.

Thank you very much for your comments and suggestions. We believe that it has helped us to improve our manuscript.
Best regards
Patricio Venegas-Aravena on behalf of the authors